# Evaluation of microalbuminuria as a prognostic indicator after a TIA or minor stroke in an outpatient setting: the prognostic role of microalbuminuria in TIA evolution (ProMOTE) study

W David Strain [1,2,3] Salim Elyas,[1,2] Nicola Wedge,[2,3] Luke Mounce [4] William Henley,[4] Martin James,[2] Angela C Shore[1,3]

[1]Diabetes and Vascular Research Centre, University of Exeter, Exeter, UK
[2]Academic Department of Healthcare for Older People, Royal Devon and Exeter NHS Foundation Trust, Exeter, UK
[3]NIHR Exeter Clinical Research Facility, University of Exeter Medical School, Exeter, UK
[4]Institute of Health Research, University of Exeter Medical School, Exeter, UK

**Correspondence to**
Dr W David Strain;
d.strain@exeter.ac.uk

## ABSTRACT

**Objective** Transient ischaemic attacks (TIA) and minor strokes are important risk factors for further vascular events. We explored the role of albumin creatinine ratio (ACR) in improving risk prediction after a first event.
**Setting** Rapid access stroke clinics in the UK.
**Participants** 2202 patients attending with TIA or minor stroke diagnosed by the attending stroke physician, able to provide a urine sample to evaluate ACR using a near-patient testing device.
**Primary and secondary outcomes** Primary outcome was major adverse cardiac events (MACE: recurrent stroke, myocardial infarction or cardiovascular death) at 90 days. The key secondary outcome was to determine whether urinary ACR could contribute to a risk prediction tool for use in a clinic setting.
**Results** 151 MACE occurred in 144 participants within 90 days. Participants with MACE had higher ACR than those without. A composite score awarding a point each for age >80 years, previous stroke/TIA and presence of microalbuminuria identified those at low risk and high risk. 90% of patients were at low risk (scoring 0 or 1). Their 90-day risk of MACE was 5.7%. Of the remaining 'high-risk' population (scoring 2 or 3) 12.4% experienced MACE over 90 days (p<0.001 compared with the low-risk population). The need for acute admission in the first 7 days was twofold elevated in the high-risk group compared with the low-risk group (3.23% vs 1.43%; p=0.05). These findings were validated in an independent historic sample.
**Conclusion** A risk score comprising age, previous stroke/TIA and microalbuminuria predicts future MACE while identifying those at low risk of a recurrent event. This tool shows promise in the risk stratification of patients to avoid the admission of low-risk patients.

## INTRODUCTION

Stroke is the second most common cause of death and a leading cause of disability worldwide.[1] Despite, or possibly because of, recent trends in reducing stroke mortality, the health and social disability burden of stroke

### Strengths and limitations of this study

► The pragmatic design of this study provides good generalisability to clinical practice.
► The predictive role of the combination of urinary albumin creatinine ratio (ACR) in combination with very basic demographics would be able to reassure the 90% of the population that there was only approximately a 1 in 20 risk of an event recurring, whereas the higher risk population had a 1 in 8 chance of a major adverse cardiac events (MACE) outcome.
► The study is limited by the availability of data which were available to the physicians at the time of the stroke appointment. This limits the ability to determine mechanistic association between ACR and MACE outcome.
► Further work is required to determine whether therapies that reduce ACR can modify the risk of subsequent MACE.

is increasing.[2] After advancing age, transient ischaemic attacks (TIA) and minor strokes are the most important risk factors for recurrent stroke and predict long-term mortality.[3 4] About 23% of patients presenting with stroke have a history of TIA in the 3 months prior to the index event.[5] This is a key population to target for secondary prevention, but these patients represent <10% of all those who present to rapid access TIA clinics.[6 7] Notably, half of all completed strokes occur within the first week after TIA or minor stroke.[5]

Accurate identification of those patients presenting to TIA clinics with TIA or minor stroke who are most at risk for future events is important to (1) intensify treatments such as giving dual antiplatelet therapy,[8 9] (2) guide the need for urgent admission to facilitate the detection and urgent surgical correction

of severe internal carotid artery (ICA) stenosis[10] and (3) reassure patients with lowest risk.

Several risk stratification tools are already used in such clinics such as ABCD2[11][12] (awarding points for *Age*, presenting *B*lood pressure, *C*linical features of unilateral weakness or aphasia, *D*uration of symptoms and *D*iabetes), California score[6] and imaging-based scoring systems.[10][13] These tools lack optimal sensitivity and specificity.[12][14] Indeed some studies suggest patients with a 'low-risk' ABCD2 score (<4/7) have similar 90-day stroke risk as patients deemed high risk with an ABCD2 score >4/7,[15] while missing up to 40% of patients with severe ICA stenosis.[16] As such, recent consensus guidelines have advised against their use.[17–19]

Increased urinary albumin excretion rate (AER) has been shown to predict incident stroke and heart failure in people with diabetes. In the general population, AER predicts cardiovascular disease and mortality post stroke independent of conventional cardiovascular risk factors such as hypertension, diabetes and smoking.[20–23] Urinary albumin creatinine ratio (ACR) is a well-recognised proxy for urinary AER, and can now be assessed using simple and inexpensive point-of-care equipment. In an earlier pilot study of 142 patients with minor stroke/TIA,[24] we identified the potential role of urinary ACR in identifying those at highest risk of a recurrent event. Although statistically significant, this study was not large enough to explore potential confounding. We therefore performed a definitive study to assess whether urinary ACR improved the risk stratification of patients presenting with TIA or minor stroke to UK acute stroke clinics.

## METHODS

Patients diagnosed with suspected minor stroke/TIA by a consultant stroke physician, attending a rapid access clinics in one of the 12 UK hospitals were recruited at the end of their clinic consultation. As these clinics are rapid access, often within the first 24 hours of symptoms, it was impossible to accurately differentiate between TIA (with symptoms <24 hours) and minor stroke in all cases. Only individuals with events that had occurred within the previous month were included in the study. Routine clinical care, including urgent revascularisation in those with severe ICA stenosis (>50% diameter stenosis by the North American Symptomatic Carotid Endarterectomy Trial method),[25] was initiated prior to enrollment. If a cardioembolic source was identified, anticoagulants were initiated according to local protocols. After written, informed consent was obtained in the clinic, demographics, including age, sex, height, weight, medical history and ABCD2 score were recorded. Times from onset of symptoms to assessment and enrollment in the study were documented. A clean specimen of urine was collected from participants in the clinic and tested with a Unistix dipstick and, if there was no indication of urinary tract infection (presence of any two of leucocytes, nitrites and/or protein), ACR was measured using a

point-of-care analyser (The Afinion AS100 Analyzer; Axis Shield, Dundee, UK). This system uses an immunometric membrane flow-through principle for albumin measurement and an enzymatic colorimetric test for creatinine quantification[26] and reports an ACR in approximately 5 min within the range 0.1–140 mg/mmol with a coefficient of variation of 4.6%–6%.[26] Participants with values of <0.1 mg/mmol were recorded as having the lowest recordable value of ACR (0.1 mg/mmol).

Participants were followed up by telephone on day 7, 30 and 90, and any history of any further vascular events, hospitalisation or death was obtained. If contact with participants was not possible initially, the presence or absence of further events was verified during additional attempts at contact or during future follow-up. If no further contact was possible with the participant (eg, in the case of significant stroke or death), medical history was collected from the next of kin and verified from the primary care physician and hospital records. All reported clinical events were adjudicated according to the standardised diagnostic criteria by a data monitoring committee including two independent stroke physicians, blinded to the ACR, by examination of clinical records if they attended hospital, or their primary care physician, or by verifying their clinical symptoms from the research records if patients did not seek further medical advice. In the event of disagreement, a third independent stroke physician acted as adjudicator. The primary outcome was to determine the utility of microalbuminuria in predicting the time to first major adverse cardiovascular event (MACE: recurrent stroke, myocardial infarction or death) within 90 days. Secondary outcomes were time to explore the predictive role of microalbuminuria on recurrent stroke, total mortality (even if these were not the first MACE), the presence of first MACE within 7 days, and the need for hospitalisation within the first 7 days. All events were adjudicated blind to urinary ACR by an independent data monitoring committee.

### Ethics statement

The study protocol was approved by the national research ethics committee (approval 14/EE/1106). All participants provided written informed consent prior to enrolment and confirmed their willingness to continue participation at each telephone consultation.

### Patient and public involvement

Patients were involved in the design and conduct of this research. During the feasibility stage, priority of the research question and methods of recruitment were informed by discussions with patients through a focus group session and three structured interviews. During the trial, two patients joined the independent trial steering committee. Once the trial has been published, participants will be informed of the results through a dedicated newsletter suitable for a non-specialist audience, led by one of the patients on the steering committee.

## Statistical analysis

Data were treated as continuous variables wherever possible to maximise power. All normally distributed data are presented as mean±SD. Skewed data were appropriately transformed and presented as geometric mean scores (with 95% CIs). Statistical significance for categorical variables was calculated using the $\chi^2$ test and the Student's t-test for continuous variables. Time to event was measured from the clinic consultation rather than the index event, in keeping with the pragmatic nature of the study. Independence of ACR (as a risk predictor) from diabetes and other measures of the ABCD2 score was assessed using logistic regression. Microalbuminuria was defined using the currently accepted clinical thresholds for microalbuminuria used in people with diabetes 3.5 mg/mol for women and 2.5 mg/mol for men. Identification of independent predictors of recurrent MACE was performed by backwards stepwise logistic regression, commencing with all information available for the complete dataset, representing all information that would routinely be available to the clinician in the clinic setting. Where multiple related measures were available (eg, systolic, diastolic and mean arterial blood pressure), the variable that accounted for the greatest degree of variance in a univariate analysis was included. Fractional polynomial regression modelling was used to identify sex-specific thresholds of ACR to define 'microalbuminuria' in the context of stroke risk stratification. Prediction of events were calculated using a Cox proportional HR with time to first MACE as the primary outcome. In keeping with the recommendations of Cupples et al[27] and Rothman,[28] the measured significance of the variables of interest is reported without adjustment for multiple testing. Statistical significance was considered at p<0.05. Statistical analysis was performed using Stata SE V14.2 (Mac version: StataCorp LLC, College Station, TX, USA).

## RESULTS

A total of 2408 patients with a diagnosis of definite or probable minor stroke/TIA were recruited; 149 participants were subsequently excluded after the diagnosis

**Table 1** Sample characteristics of total population and population stratified by occurrence of recurrent vascular events or death (MACE) at 90 days

| Characteristic | Total population | No event by 90 days | Event at 90 days |
|---|---|---|---|
| N | 2202 | 2058 | 144 |
| Days from event to clinic† | 3 (1–6) | 3 (1–6) | 2 (1–4) |
| Age (years) | 70.9 (11.7) | 70.9 (11.6) | 72.2 (12.2) |
| Sex (% male) | 64.8 | 64.7 | 67.9 |
| Height (m) | 1.70 (0.10) | 1.70 (0.10) | 1.71 (0.9) |
| Weight (kg) | 79.8 (17.1) | 79.9 (17.0) | 79.7 (17.7) |
| BMI (kg/m$^2$) | 27.6 (5.2) | 27.6 (5.2) | 27.2 (4.9) |
| Blood pressure | | | |
| Systolic (mm Hg) | 144.1 (20.8) | 144.2 (20.9) | 145.7 (21.7) |
| Diastolic (mm Hg) | 81.8 (12.6) | 81.7 (12.6) | 83.8 (13.1) |
| Total cholesterol | (n=1466) | (n=1374) | (n=92) |
| (mmol/L)‡ | 5.02 (2.57) 194 | 5.03 (2.63) | 4.75 (1.24) |
| (mg/dl)‡ | 194(99) | 195 (102) | 183 (48) |
| Diabetes (%) | 15.7 | 15.9 | 12.1 |
| Previous stroke (%) | 4.0 | 3.5 | 10.7* |
| ABCD2 | | | |
| Mean score out of 7 | 4.0 (1.4) | 4.0 (1.4) | 4.1 (1.4) |
| Proportion >4 (%) 'High risk' | 62.5 | 62.3 | 65.0 |
| ACR (mg/mmol)§ | 1.79 (0.9–2.9) | 1.75 (1.69–1.83) | 2.27 (1.92–2.68)** |
| ACR (mg/g)§ | 15.8 (8.0–25.6) | 15.5 (14.9–16.2) | 20.1 (17.0–23.7)** |

*p<0.0001 compared with those with no events; **p=0.002 compared with those with no events.
†Median (IQR).
‡These data were collected from patient records where clinician had requested the test and therefore only available in a subset of the population.
§Geometric mean (95% CI).
ABCD2, Age, presenting Blood pressure, Clinical features of unilateral weakness or aphasia, Duration of symptoms and Diabetes; ACR, albumin creatinine ratio; BMI, body mass index; MACE, major adverse cardiac events.

was revised to that of a stroke mimic, 8 withdrew consent and a further 49 (2%) were excluded due to intercurrent urinary tract infection rendering the assessment of ACR invalid. No patient was lost to follow-up. Baseline characteristics are presented in table 1. Of the 2202 included in the final analysis, most were male (64.9%). All patients were commenced on secondary prevention with appropriate antiplatelet or anticoagulant therapy, and an appropriately dosed statin in accordance with contemporaneous national guidelines.

Over 90 days, 151 primary outcome events (MACE) occurred in 144 participants (6.7% of patients) including 8 cardiovascular deaths. All MACE were atherosclerotics with no haemorrhagic events occurring in the 3 months. There were also eight non-cardiovascular deaths. Within 7 days, there were 38 MACE in 36 participants. Those with MACE were more likely to have had previous stroke/TIA, however, with this exception, there were no significant differences in routinely collected clinic data (table 1). There was no difference in mean ABCD2 score or the proportion of participants with a 'high-risk' score (4/7 or more) between those who did and did not have events by 90 days. When looking exclusively at the 1374 participants identified with a 'high-risk' ABCD2 score (63% of the study sample; 93% male), a high score did not distinguish those at high risk of MACE compared with those with an ABCD2 <4 (HR 1.15 (0.82–1.63); p=0.4), nor did a low risk score exclude an event, with only 37.7% of people who have not experienced MACE outcome scoring <4. Point-of-care ACR was higher in those participants with a primary outcome event at 90 days compared with those without (2.27 (1.92–2.68) vs 1.75 (1.68–1.83) mg/mmol, respectively; p=0.002). This distinction remained after adjustment for age and sex (adjusted ACR 2.40 (2.04–2.83) vs 1.88 (1.80–1.96), p=0.004; figure 1).

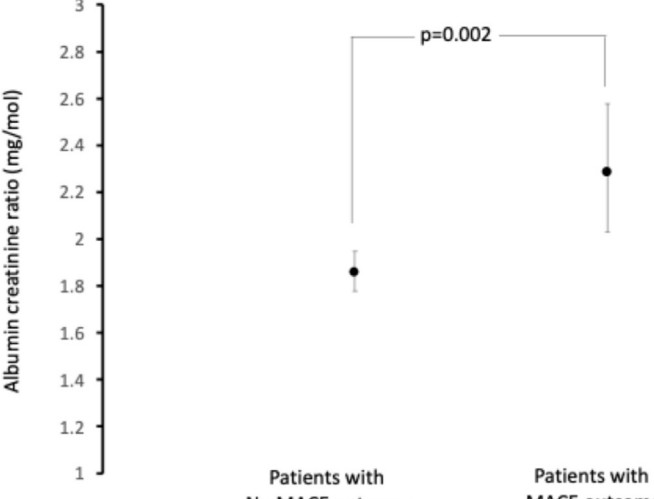

**Figure 1** Age and sex-adjusted albumin creatinine ratio (mean±95% CI) stratified by the occurrence of MACE in the first 90 days post initial event. MACE, major adverse cardiovascular events such as recurrent cerebrovascular event, myocardial infarction or cardiovascular death.

### Evaluation of the prognostic role of microalbuminuria alone

Fractional polynomial regression modelling identified the sex-specific threshold of risk for ACR aligned with the currently accepted thresholds for microalbuminuria used in people with diabetes of 2.5 mg/mmol for men and 3.5 mg/mol for women. These thresholds identified 562 participants (25.4% of the study sample; 75% male) with microalbuminuria. These participants were more likely than those with a low ACR to experience a primary outcome event (HR 1.66 (95% CI 1.19 to 2.34); p=0.003) and had a greater than fivefold increase in 90-day mortality (HR 5.44 (1.86–15.90); p=0.0003). Alone, however, microalbuminuria did not have sufficient positive or negative predictive value to justify its clinical utility (positive predictive value (PPV): 9.1%, specificity: 75.4%).

### Generating a composite risk score for 90-day risk stratification

In a stepwise multivariate model, the only independent predictors of MACE were elevated ACR, age >80 years and history of stroke or TIA. Similar variance within the model was explained microalbuminuria (ie, an ACR >2.5 mg/mol and 3.5 mg/mol, respectively, in men and women) as being older than 80 years of age and history of stroke/TIA, so they were afforded equal weighting. This generated a 4-point scale between 0 and 3 (termed the 'APA' score representing age, prior stroke or TIA and elevated ACR), with the lowest risk participants having a score of 0 (online supplemental table 1). There was a sequential increase in risk with increasing score such that those with a score of 0 having a 90-day risk of 4.86% rising to a 30% risk for those with the highest score (online supplemental table 2). The population was then divided into a low-risk group scoring 0 or 1 or high-risk scoring 2 or 3. A total of 1985 (90.1%) of participants were identified as low risk compared with 217 (9.9%) in the high-risk group. When comparing the PPV sensitivity and specificity of the APA score compared with the previously available parameters of age and previous stroke demonstrated a clinically meaningful improvement over each measure on its own (online supplemental table 3A). Compared with using previous stroke alone, APA score had a similar PPV (12.4% for APA score vs 11.9% for previous stroke) and specificity (91.2% vs 96.4%, respectively), however superior sensitivity (19.3% vs 7.1%, respectively). Sensitivity was improved using a composite of previous stroke and being above the age of 80 years; however, this was at the expense of a reduction of PPV to 8.4%.

The high-risk group was older, with higher systolic blood pressure and pulse pressure compared with the low-risk group (table 2). The 90-day risk of MACE outcome was 5.7% in the low-risk group versus 12.4% in the high-risk group (p<0.001), translating to a HR of 2.12 (95% CI 1.38 to 3.25) for the high-risk score compared with low-risk group (p<0.001; figure 2). This was predominantly driven by an increase in recurrent stroke/TIA (3.3% vs 8.8%; HR 2.67 (1.58–4.51); p<0.001). This difference was apparent within

**Table 2** Occurrence of events by APA risk score group (mean±SD for continuous variable, p for ANOVA; % for categorical data, p for χ²)

|  | Low risk | High risk | P value |
|---|---|---|---|
| n | 1985 | 217 |  |
| Sex (% male) | 64.5 | 68.2 | 0.2 |
| Age (years) | 69.6 (11.3) | 84.0 (5.0) | <0.0001 |
| Blood pressure (mm Hg) |  |  |  |
| Systolic | 143 (20.4) | 150.5 (23.3) | <0.0001 |
| Diastolic | 82 (12.3) | 79 (11.6) | 0.0007 |
| Weight (kg) | 80.3 (17.1) | 75.1 (16.2) | <0.0001 |
| Diabetes | 15.3 | 18.9 | 0.2 |
| MACE at 90 days (%) |  |  |  |
| All events | 5.7 | 12.4 | <0.0001 |
| Cerebrovascular event | 3.2 | 8.8 | <0.0001 |
| Cardiovascular mortality | 0.1 | 1.4 | <0.001 |
| All cause mortality | 0.4 | 2.3 | 0.001 |
| MACE at 7 days (%) | 1.46 | 3.23 | 0.05 |

ACR, albumin creatinine ratio; APA, age, prior stroke or TIA and elevated ACR; MACE, major adverse cardiac events; TIA, transient ischaemic attacks.

7 days of the clinic, such that those with a high APA score had a 2.2-fold increased risk of a recurrent event.

One thousand and forty-nine participants were reviewed within 48 hours of their index event. For these individuals, the predictive role of the APA score was numerically superior (HR 2.84, 95% CI 1.32 to 6.14; p=0.008) than those who were seen with a longer delay. There was no significant interaction between time from event to appointment and predictive role of the APA score.

### Validation of APA score on a historic sample

To validate the APA score, we applied the tool to the individuals who participated in the pilot study but did

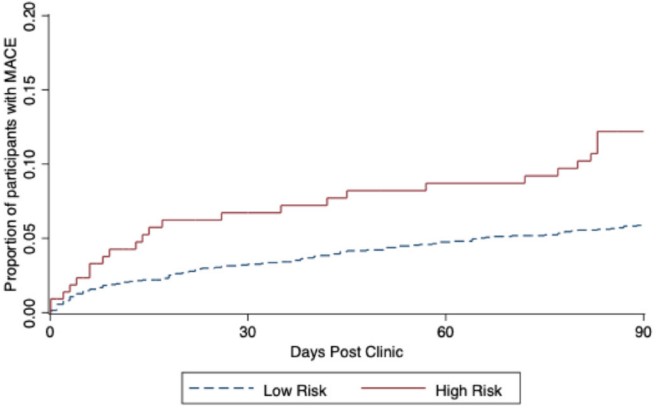

**Figure 2** Time to first MACE stratified by age, previous cerebrovascular event, high ACR risk score. Event rate in high-risk group (n=217) 12.44% versus low-risk group (n=1985) 5.69%, HR 2.12 (1.38–3.25) p<0.001. ACR, albumin creatinine ratio; MACE, major adverse cardiac events.

not contribute their data to this dataset. Details of this pilot study have been published elsewhere.[24] In brief, the pilot study recruited 139 participants over 9 months from one of the centres involved in the definitive study. In this population, there were 13 recurrent events (9.35%) within the 90 days, 9 of which occurred in the first 7 days after attendance to the stroke clinic (table 3). The APA risk score replicated a similar distribution of number of participants at low risk (73%) and high risk (27%) as in the larger study. Compared with the component parts, the APA score had superior positive predictive value (24.4% vs 18.2% for previous stroke or 13.3% for previous stroke and being aged >80 years) and superior sensitivity (76.9% vs 46.2% vs 15.4%, respectively; online supplemental table 3B). At 90 days, 5.9% of the low APA group had recurrent events compared with 18.4% of the in the high APA group (p=0.02). Again, within the first 7 days this was apparent with only 1.9% of the low APA group experiencing the primary outcome, compared with 13.2% in the high APA group (p=0.04).

### DISCUSSION

In this large UK multicentre prospective study with blinded outcome adjudication, we have demonstrated the utility of an elevated ACR, measured using a simple point-of-care analyser, in combination with simple clinical data (age and history of stroke or TIA) in the prediction of MACE and death. This differentiates between low-risk and high-risk individuals. Specifically, >90% of individuals were identified as low risk with<1.5% risk of a MACE or death over the next 7 days, increasing to 5.7% by 90 days, whereas 9.9% of the population were identified as high

**Table 3** Occurrence of events by APA risk score group in independent validation population (mean±SD for continuous variable, p for ANOVA; % for categorical data, for $\chi^2$)

|  | Low risk | High risk | P value |
|---|---|---|---|
| n | 101 | 38 |  |
| Sex (% male) | 51.5 | 71.5 | 0.04 |
| Age (years) | 72.6 (10.7) | 81.6 (6.1) | 0.001 |
| Blood pressure (mmHg) |  |  |  |
| Systolic | 142.6 (20.2) | 147.3 (26.6) | 0.3 |
| Diastolic | 77.7 (9.8) | 77.4 (10.8) | 0.9 |
| Weight (kg) | 76.2 (16.4) | 75.0 (15.4) | 0.7 |
| Recurrent event at 90 days (%) |  |  |  |
| MACE | 5.9 | 18.4 | 0.02 |
| Recurrent event at 7 days (%) |  |  |  |
| MACE* | 1.9 | 13.2 | 0.04 |

*MACE include recurrent stroke or TIA, myocardial infarction or cardiovascular death.
ACR, albumin creatinine ratio; APA, age, prior stroke or TIA and elevated ACR; MACE, major adverse cardiac events; TIA, transient ischaemic attacks.

risk with a 12.4% event rate over 90 days. Although only a small number, our findings were validated in a separate historic population who participated in the pilot study.

The utility of APA scoring may go beyond simply guiding conversations regarding prognosis in a clinic setting. The Platelet-Oriented Inhibition in New TIA and Minor Ischaemic Stroke (POINT) Trial[8] and (Clopidogrel in High-Risk Patients with Acute Nondisabling Cerebrovascular Events) CHANCE trial[9] demonstrated the benefit of dual antiplatelet therapy in the first 21 days after high-risk TIA. There was, however, a 0.5% absolute increase in major intracranial haemorrhage. We would propose that an assessment of the APA score could be evaluated as a stratification tool to determine whether it predicts those who benefit the most. In those with a low-risk APA score, even the 25% relative risk reduction in the POINT trial would result in only a marginally beneficial risk:benefit ratio, assuming that microalbuminuria is a predictor of thromboembolic disease processes. Those with a high APA score, however, would likely benefit from dual antiplatelet therapy with a number need to treat of <30 to prevent an event. This suggestion, however, does require confirmation in an independent study. The APA score also gives additional information that is of use in the more acute setting. Over the first 7 days, a low-risk APA score had a negative predictive value of >98.5% for the risk of MACE—a number very similar to the predictive effect of troponin when evaluating suspected cardiac chest pain.[29]

When considering the populations identified by the APA score, there are similarities to the ABCD2 score. Indeed, the demographics of the high-risk group were older, with higher blood pressure and a trend towards more diabetes. We would suggest, that microalbuminuria represents vascular susceptibility to the adverse consequences of hypertension and diabetes rather than simply an indication of the presence of hypertension and/or diabetes in general. This would explain why the APA score predicted further MACE and total mortality at 7 and 90 days, whereas in our large sample the ABCD2 score did not.

Elevated ACR is a recognised marker of generalised endothelial and microvascular dysfunction. The increased filtration of albumin through the renal glomerular filtration barrier is thought to be due to changes in the chemical and physical properties of this endothelial barrier and its glycocalyx.[30–32] The mechanism explaining the association between microalbuminuria and incident cardiovascular events is thought to depend on its role as a marker of systemically increased vascular permeability and altered homeostasis, coagulation and endothelial function.[33–38] However, after acute ischaemic events such as myocardial infarction, or as demonstrated here TIA or minor stroke, it is likely that elevated urinary ACR is, at least in part, dependent on the systemic inflammatory response to the original insult.

Although unlikely to be mechanistically involved in the progression of disease, the question as to whether it is a good surrogate of therapeutic effect remains. Improving systemic inflammation, such as through the pleiotropic effect of statins, has been associated with simultaneous improvements in urinary albumin excretion and cardiovascular event rates in patients with elevated high sensitivity C reactive protein.[39–41] Further studies are needed to determine whether microalbuminuria after acute stroke represents a therapeutic target amenable to treatment, and if reductions in microalbuminuria are associated with reductions in the vascular events they predict.

### Limitations and strengths of this study

Recurrent event rate was lower than anticipated from our pilot study and contemporaneous clinical trials such

as CHANCE (11.7% in the control arm),[9] although are comparable with the recently reported randomised controlled POINT trial (6%).[8] In usual practice it is usually 24–48 hours between the index event and clinic attendance. Given that data from previous studies has demonstrated the highest risk occurs in the first 48 hours, it is likely that some events will have occurred between the index event and attendance in the clinic in our study.[42] The pragmatic design of the study, however, is also a strength given that the predictive role highlights those that may benefit from dual antiplatelet therapy or hospitalisation when the patient is seen in the acute stroke clinic.

There is no consensus on the gold standard for diagnosing TIA in a rapid access stroke clinic.[43] In order to maintain the generalisability of the study, all attendees of the stroke clinic that were diagnosed with a probable or definite TIA were invited to participate. Only a small number of patients were subsequently withdrawn as being stroke mimics. Outcome events were rigorously adjudicated against published diagnostic criteria to ensure consistency and reliability.

Finally, the use of a single urinary ACR sample obtained during the clinic assessment using a near patient testing kit is not as robust as a sample tested in a central laboratory. Although there is significant diurnal variability in urinary AER within individuals, potentially limiting the scientific or mechanistic merit of the study compared with studies using overnight or 24-hour urine collection, the pragmatic design makes this study more applicable to general clinic populations. Furthermore, the relatively low cost of the near patient test kit makes it accessible in most emergency settings, in primary and secondary care.

## CONCLUSION

We have demonstrated for the first time the potential utility of a single point-of-care test of urinary ACR in patients presenting with TIA and minor stroke as a prediction tool to assist triaging patients. When used in combination with patient age and history of stroke or TIA, this generated a risk stratification score (the APA score: age >80 years, previous stroke/TIA and elevated ACR) that could reliably identify the large proportion of patients with a low risk of recurrent vascular event over the next 90 days. The value of ACR that is most predictive is aligned with the conventional definition of microalbuminuria used in diabetes, suggesting that in clinical practice the existing technology for evaluating microalbuminuria including urine dipsticks may be used. A high APA score was associated with a doubling of risk of MACE outcome and a fivefold increase in mortality. This APA score was validated in an independent population. We believe the APA score may represent a practical tool for clinicians engaged in the acute ambulatory assessment of TIA and minor stroke, to assist in the assessment of risk:benefit ratio for the use dual antiplatelet therapy or to admit for more urgent investigation. Further work is required to determine whether the increase in ACR represents a therapeutic target or solely a prognostic indicator.

**Acknowledgements** WDS is supported by the NIHR Exeter Clinical Research Facility. WDS and MAJ is supported by the NIHR SW Peninsula Applied Research Collaboration.

**Contributors** WDS and ACS conceived and designed the study, attained funding, were involved in the management of the trial. WDS produced the first draft of the manuscript. SE, NW and MJ contributed to the design and management of the trial. LM and WH provided the statistical support for the management of the trial and the analysis of the outcomes. All authors have reviewed the final manuscript prior to submission.

**Funding** This study was funded by a NIHR Research for Patient Benefit grant (PB-PG-1112-29069) and supported by the NIHR Exeter Clinical Research Facility.

**Disclaimer** This manuscript does not necessarily reflect the views of the NIHR, the Exeter Clinical Research Facility, the NHS or the UK Department of Health.

**Competing interests** None declared.

**Patient consent for publication** Not required.

**Provenance and peer review** Not commissioned; externally peer reviewed.

**Data availability statement** Data are available on reasonable request to the corresponding author.

**ORCID iDs**
W David Strain http://orcid.org/0000-0002-6826-418X
Luke Mounce http://orcid.org/0000-0002-6089-0661

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
