## [Reviewer comments · BMJ Open]

ARTICLE DETAILS

TITLE (PROVISIONAL)	The evaluation of microalbuminuria as a prognostic indicator after a TIA or minor stroke in an outpatient setting: The Prognostic Role of Microalbuminuria in TIA Evolution (ProMOTE) Study
AUTHORS	Strain, W; Elyas, Salim; Wedge, Nicola; Mounce, Luke; Henley, W; James, Martin; Shore, Angela

VERSION 1 – REVIEW

REVIEWER	Sander, Dirk Technische Universität München
REVIEW RETURNED	12-Oct-2020

GENERAL COMMENTS	The authors evaluated the role of albumin-creatinine ratio (ACR) in improving risk prediction after a first event. 2202 patients with TIA or minor stroke were included in the prospective and multi-center study. 151 major adverse cardiovascular events (MACE) occurred within 90 days. A composite risk score (APA) awarding a point each for age > 80 years, previous Stroke/TIA and presence of microalbuminuria identified patients with low (0 or 1 point) and high (2 or 3 points) risk with a 90 day MACE of 5,7% and 12,4% respectively. The need for acute admission during the first 7 days was two-fold elevated in the high-risk as compared to the low-risk group (3,23% vs. 1,43%; p=0,05). The authors concluded, that this risk score shows promise in the risk stratification of patients to avoid the admission of low risk patients whilst assigning intensive interventions therapy to those at highest risk. The topic of the study is of potential interest for the readers of „BMJ Open“. The data are clearly presented and the study design is described in detail. The statistics needs some clarification. The limitations have been mentioned and discussed. The findings are of clinical importance because risk stratification after TIA or minor stroke is important to optimize the intensity of early secondary prevention. Specific comments: * The authors performed a stepwise multivariate analysis to evaluate independent predictors of MACE and stated that only ACR, age and history of previous stroke or TIA remained in the model. The authors should describe the method and results of this analysis in more detail. Are they used forward selection or backward elimination or a bidirectional elimination approach? Which variables were tested with the procedure? Please give some data regarding model accuracy. I suggest to present the results of this important regression analysis in a separate table.
--

	* In table 1 the value for total cholesterol in mg/dl for the whole population is missing. Table 2 could be omitted, as it is explained already in the text. * The authors stated the the risk score used is useful for risk stratification and that ACR is simple and easy to measure. However, it would be helpful to get some idea how ACR improves risk stratification as compared to a score based only on the clinical variables, e.g. age > 80, history of stroke or TIA? Is there a clinical manful increase in sensitivity, specificity, PPV, NPV if ACR is added to the score? * One method to evaluate clinical usefulness of a parameter is to calculate the AUC or net benefit. * Several updated guidelines even recommend dual anti-platelet therapy for TIA or minor stroke for 10 to 21 days based on the POINT and CHANCE data. Therefore the authors should discuss their suggestion to select patients for dual anti-platelet therapy based on the APA score in the light of these guidelines and should state more clearly, that a controlled trial is necessary to test the usefulness of this score for treatment desicions.
--	--

REVIEWER	Ihle-Hansen, Haakon Vestre Viken HF, Department of medical research
REVIEW RETURNED	27-Dec-2020

GENERAL COMMENTS	Risk stratification scores designed for daily routine is in demand, and ACR is a promising risk marker. Hence, Strain and coworkers address an important and interesting research question. The study has a suffices size, and the pragmatic design is a strength. However, I have some concerns: 1) The first sections of the introduction contain several falls statements (e.g. "stroke is the second most common cause of death", "TIA and minor stroke are the most important risk factors for recurrent stroke") 2) There are some methodological concerns: a. At the end of the methods section, the authors state; "main primary outcome was time to first major adverse cardiovascular event within 90 days and secondary outcomes were time recurrent stroke, total mortality, the precedence of first MACE and re-hospitalisation". However, the results do not reflect the primary and secondary outcomes. b. The ratio TIA/minor stroke is not reported. c. How was a minor stroke and TIA defined? d. Were patients referred to CEA counted as re-hospitalisation? ("within 7 days there were 38 MACE in 36 participants and further 76 event that required hospitalization, predominantly carotid endarterectomy"). 3) Several key measurements are lacking, such as in which time window was the study conducted. 4) In the results section, the authors present some results which are not supported by the data presented (e.g. Those with MACE were more likely to have had previous stroke/TIA, but were otherwise similar to those without recurrence (table 1)") 5) The manuscript is not to the point written. Sentences like "such as giving dual anti-platelet therapy with aspirin and clopidogrel in combination (Introduction, second paragraph)" and "Routine clinical care would include initiation of single antiplatelet, cholesterol-lowering and anthihypertensive treatment as soon as possible ... (Methods, first section)" are partially incorrect and not required for the content of the manuscript.
--

VERSION 1 – AUTHOR RESPONSE

Reviewer: 1

Dr. Dirk Sander, Technische Universität München Comments to the Author:

The authors evaluated the role of albumin-creatinine ratio (ACR) in improving risk prediction after a first event. 2202 patients with TIA or minor stroke were included in the prospective and multi-center study. 151 major adverse cardiovascular events (MACE) occurred within 90 days. A composite risk score (APA) awarding a point each for age > 80 years, previous Stroke/TIA and presence of microalbuminuria identified patients with low (0 or 1 point) and high (2 or 3 points) risk with a 90 day MACE of 5,7% and 12,4% respectively. The need for acute admission during the first 7 days was two-fold elevated in the high-risk as compared to the low-risk group (3,23% vs. 1,43%; $p=0,05$). The authors concluded, that this risk score shows promise in the risk stratification of patients to avoid the admission of low risk patients whilst assigning intensive interventions therapy to those at highest risk.

The topic of the study is of potential interest for the readers of „BMJ Open“. The data are clearly presented and the study design is described in detail. The statistics needs some clarification. The limitations have been mentioned and discussed. The findings are of clinical importance because risk stratification after TIA or minor stroke is important to optimize the intensity of early secondary prevention.

Thank you for these comments.

Specific comments:

- * The authors performed a stepwise multivariate analysis to evaluate independent predictors of MACE and stated that only ACR, age and history of previous stroke or TIA remained in the model. The authors should describe the method and results of this analysis in more detail. Are they used forward selection or backward elimination or a bidirectional elimination approach? Which variables were tested with the procedure? Please give some data regarding model accuracy.

Thank you for this comment. We used backwards stepwise logistic regression starting with all variables that would routinely be available to the clinician in clinic. Where multiple different measures of the same parameter are available (e.g. blood pressure – Systolic BP, Diastolic BP, Mean arterial Pressure) the variable that accounted for the greatest amount of the variance in a univariate analysis was considered. Alternative models were examined using threshold values (i.e. age > 80 years, conventional measure of microalbuminuria) vs using the data as a continuous variable. Interestingly, as a continuous variable, age only had limited predictive value, however the threshold of 80 marked a significant point in prediction, whereas microalbuminuria (log transformed) provided a good association whether used continuously or dichotomously.

We have now included more information in the statistical analysis and more details of the outcomes in the results. – Page 10, Para 1

- * In table 1 the value for total cholesterol in mg/dl for the whole population is missing.

We apologise for this, it has now been included. – Table 1

Table 2 could be omitted, as it is explained already in the text.

Whilst we agree that this is described in the text in some detail, Table 2 represents the simplified message of the study, and we believe this is the table that will most commonly be reproduced afterwards when describing the tool. We would be happy to move this to a supplementary table at the discretion of the editor

* The authors stated the the risk score used is useful for risk stratification and that ACR is simple and easy to measure. However, it would be helpful to get some idea how ACR improves risk stratification as compared to a score based only on the clinical variables, e.g. age > 80, history of stroke or TIA? Is there a clinical manful increase in sensitivity, specificity, PPV, NPV if ACR is added to the score?

Thank you for this comment, highlighting possibly the most important element of the study For your information The APA score presents the best compromise across the board

The predictive value of being over the age of 80

		Aged > 80 years	
		Yes	No
Recurrent MACE event	45		95
No further Event	522		1540

PPV	7.9%
NPV	94.2%
Sensitivity	32.1%
Specificity	74.7%
p=0.07	

Predictive value of previous stroke

		Previous Stroke	
		Yes	No
Recurrent MACE event	10		130
No further Event	74		1999

PPV	11.9047619
NPV	93.8938469
Sensitivity	7.14285714
Specificity	96.4302943
p<0.001	

Predictive value of previous stroke and being over the age of 80 Age>80 & Previous Stroke score

		Yes	No
Recurrent MACE event	53		87

No further Event	576	1486
	PPV	8.4%
	NPV	94.5%
	Sensitivity	37.9%
	Specificity	72.1%
	p<0.001	

Predictive value of APA score

	APA score	
	High Risk	Low Risk
Recurrent MACE event	27	113
No further Event	190	1972

PPV	12.4%
NPV	94.6%
Sensitivity	19.3%
Specificity	91.2%
	p<0.001

Very similar numbers were seen in the validation population here in table form for convenience

	PPV	NPV	Sensitivity	Specificity		
Aged >80 years			9.8%	91.3%	38.5%	64.6%
Previous Stroke			18.2%	93.6%	46.2%	79.2%
Age >80 years and Previous Stroke			13.3%	91.4%	15.4%	90.0%
APA score ≥2			24.4%	96.9%	76.9%	95.4%

We have presented these data in supplementary table 2 and we have referenced them in the text – Page 12 para 2 & page 13 para 3

* Several updated guidelines even recommend dual anti-platelet therapy for TIA or minor stroke for 10 to 21 days based on the POINT and CHANCE data. Therefore the authors should discuss their suggestion to select patients for dual anti-platelet therapy based on the APA score in the light of these guidelines and should state more clearly, that a controlled trial is necessary to test the usefulness of this score for treatment decisions.

We agree entirely with this suggestion, and are currently planning the follow up study. We have highlighted that this is a hypothesis that requires testing in the text. Page 15 para 1

Reviewer: 2

Dr. Haakon Ihle-Hansen, Vestre Viken HF, University of Oslo Comments to the Author:

Risk stratification scores designed for daily routine is in demand, and ACR is a promising risk marker. Hence, Strain and coworkers address an important and interesting research question. The study has a suffices size, and the pragmatic design is a strength.

However, I have some concerns:

1) The first sections of the introduction contain several false statements (e.g. “stroke is the second most common cause of death”, “TIA and minor stroke are the most important risk factors for recurrent stroke”)

Thank you for this comment. We acknowledge that advancing age is the most important risk factor, with previous stroke or TIA being second to that. With regard to the statement about stroke being the second most common cause of death. We believe this still to be true. In the most recent global burden of disease study (2017) Stroke still ranked the second most common cause of death (with 5.5m deaths per annum). Even in these strange COVID times, that still dwarfs the global death toll from COVID-19. WE would happily correct this if the reviewer is aware of as yet unpublished data that suggests interventions have reduced the impact of stroke on global mortality. Page 6 para 1

2) There are some methodological concerns:

a. At the end of the methods section, the authors state; “main primary outcome was time to first major adverse cardiovascular event within 90 days and secondary outcomes were time recurrent stroke, total mortality, the precedence of first MACE and re-hospitalisation”.

However, the results do not reflect the primary and secondary outcomes.

You are absolutely correct. The primary outcome, as the title suggests, was to determine the predictive role of microalbuminuria on those events. Forgive our oversight within the text, which we have corrected. Page 9 para 1

b. The ratio TIA/minor stroke is not reported. & c. How was a minor stroke and TIA defined? This is correct due to the pragmatic study design this study was only for those seen in outpatient clinics with minor impairments in function (LACS strokes with no/minimal residual deficits) or with TIAs (deficits lasting less than 24 hours). Often clinic appointments were within the first 24-hour period therefore it was impossible to determine whether these were truly TIAs or Minor strokes. Therefore, we have not reported the distinction. We have, clarified this in the methods Page 7 para 2

d. Were patients referred to CEA counted as re-hospitalisation? (“within 7 days there were 38 MACE in 36 participants and further 76 event that required hospitalization, predominantly carotid endarterectomy”).

Patients referred for CEA were not counted as MACE events, although they were hospitalised We have removed this observation from the manuscript. The predictive role of microalbuminuria on those patients will be evaluated in a separate analysis. Deletion Page 11 para 1

3) Several key measurements are lacking, such as in which time window was the study conducted.

This has been clarified Page 7 para 2

4) In the results section, the authors present some results which are not supported by the data presented (e.g. Those with MACE were more likely to have had previous stroke/TIA, but were otherwise similar to those without recurrence (table 1))”

We apologise. This was shorthand for there was no significant difference in the routinely collected data between those with and without recurrent events apart from a past history of stroke. We have clarified this in the text. Page 11 para 2

5) The manuscript is not to the point written. Sentences like “such as giving dual antiplatelet therapy with aspirin and clopidogrel in combination (Introduction, second paragraph)” and “Routine clinical care would include initiation of single antiplatelet, cholesterol-lowering and antihypertensive treatment as soon as possible ... (Methods, first section)” are partially incorrect and not required for

the content of the manuscript. The text has been modified and these unnecessary additions have been deleted. Multiple deletions

Reviewer: 1

Competing interests of Reviewer: None declared

Reviewer: 2

Competing interests of Reviewer: None declared

VERSION 2 – REVIEW

REVIEWER	Sander, Dirk Technische Universität München
REVIEW RETURNED	12-Apr-2021

GENERAL COMMENTS	The authors responded sufficiently to my questions and suggestions. I agree to move Table 2 to a supplementary table.
---

REVIEWER	Ihle-Hansen, Haakon Vestre Viken HF, Department of medical research
REVIEW RETURNED	16-Apr-2021

GENERAL COMMENTS	In this study, Strain and coworkers have explored the predictive value of microalbuminuria after TIA and minor stroke. The study well pragmatic designed and with a respectable number of participants. The high participation rate increases the generalizability of the results. As the author correctly points out, predictive risk models after TIA and minor stroke are in demand. The aims and the results of the study are of interest, but I think the authors should concentrate on data, and leave the interpretation to the readers. To clarify: The study indicates that albumin creatinine ratio (ACR) is an independent predictor of a major adverse cardiovascular event within 90 days. A composite risk score was generated, to classify the subject as either high or low risk. The high-risk group had HR of approx 2 compared to the low-risk group. Based on these results the authors state “This tool shows promise in the risk stratification of patients to avoid the admission of low-risk patients whilst assigning intensive interventions such as dual antiplatelet therapy to those at highest risk”. The presented data do not support this statement. First of all, approx. 80% of the MACE events occurred in the low-risk group. Prioritize only on the high-risk group for intensive intervention strategy is not acceptable. Secondly, the outcome measure MACE does not differentiate between ischemic or hemorrhagic events. one should therefore be careful about commenting on/recommending intensified antithrombotic treatment in the high-risk group. I additionally have some minor comments:  1. Page 7, line 6. Stroke is not the second most common cause of death. 2. The risk of recurrence is greatest within the first weeks after the index event. The authors report number. of MACE within 7 days,
---

	but has left out the time from TIA/minor stroke to admission. Time from index event to study inclusion is of great importance for the results. 3. Table 1. Since the two groups (event/no event) are different, I would prefer that the authors report the p values
--	---

VERSION 2 – AUTHOR RESPONSE

Reviewer: 1

Dr. Dirk Sander, Technische Universität München

Comments to the Author:

The authors responded sufficiently to my questions and suggestions. I agree to move Table 2 to a supplementary table.

Thank you for this

Reviewer: 2

Dr. Haakon Ihle-Hansen, Vestre Viken HF, University of Oslo

Comments to the Author:

In this study, Strain and coworkers have explored the predictive value of microalbuminuria after TIA and minor stroke. The study well pragmatic designed and with a respectable number of participants. The high participation rate increases the generalizability of the results. As the author correctly points out, predictive risk models after TIA and minor stroke are in demand. The aims and the results of the study are of interest, but I think the authors should concentrate on data, and leave the interpretation to the readers.

To clarify:

The study indicates that albumin creatinine ratio (ACR) is an independent predictor of a major adverse cardiovascular event within 90 days. A composite risk score was generated, to classify the subject as either high or low risk. The high-risk group had HR of approx 2 compared to the low-risk group.

Based on these results the authors state “This tool shows promise in the risk stratification of patients to avoid the admission of low-risk patients whilst assigning intensive interventions such as dual antiplatelet therapy to those at highest risk”. The presented data do not support this statement.

First of all, approx. 80% of the MACE events occurred in the low-risk group. Prioritize only on the high-risk group for intensive intervention strategy is not acceptable.

We accept this criticism and will leave it with the reader to interpret. Indeed, as microalbuminuria is not usually associated with thrombotic events this may be an indicator of unmet risk. We have included a caveat to this effect and have highlighted the need for further research into this.

Secondly, the outcome measure MACE does not differentiate between ischemic or hemorrhagic events. one should therefore be careful about commenting on/recommending intensified antithrombotic treatment in the high-risk group.

Apologies for this oversight – All of the events were atherosclerotic thrombo-embolic events and there were no haemorrhagic events in the first 3 months. We have clarified this in the results section

I additionally have some minor comments:

1. Page 7, line 6. Stroke is not the second most common cause of death.

This has previously been highlighted by another reviewer, however according to the World Health Organisation and the authors quoted, Stroke remains the second leading cause of death worldwide. <https://www.who.int/news-room/fact-sheets/detail/the-top-10-causes-of-death>

2. The risk of recurrence is greatest within the first weeks after the index event. The authors report number. of MACE within 7 days, but has left out the time from TIA/minor stroke to admission. Time from index event to study inclusion is of great importance for the results.

We thank you for this comment, and have done sensitivity analyses for the purpose of validation and future studies which does demonstrate that time from initial event gives a better prognostic result, after adjustment of recorded ACR for time since event. In practice, given that the majority of patients were seen within 48 hours of the initial even the improvement was only marginal. (<3%). However, pragmatically, the tools to adjust for time since event would not be available to a clinician at the time therefore we have elected to present the T0 as the time the patient was reviewed in clinic.

3. Table 1. Since the two groups (event/no event) are different, I would prefer that the authors report the p values

All of the variables are statistically similar with the exception of previous stroke, (p<0.001) and ACR = 0.002 as highlighted with the asterixis. There were no other significant differences in this. I would be happy to include the values at the discretion of the editor, however they are all in the region of 0.4 to 0.9.

Reviewer: 1

Competing interests of Reviewer: None

Reviewer: 2

Competing interests of Reviewer: No competing interests

VERSION 3 – REVIEW

REVIEWER	Ihle-Hansen, Haakon Vestre Viken HF, Department of medical research
REVIEW RETURNED	09-Jun-2021

GENERAL COMMENTS	The ProMOTE study has a pragmatic design, is nicely conducted, and has an interesting and relevant research question. The authors responded sufficiently to my questions. However, is still have some minor concerns: 1) In the conclusion section of the abstract, the authors still claim that ACR shows promise in the risk stratification of patients to avoid the admission of low-risk patients whilst assigning intensive interventions such as dual antiplatelet therapy to those at the highest risk. The ACR is indeed a promising tool for risk stratification, but the clinical implication of the results is uncertain. Hence, the last part of the sentence should in my opinion be removed. 2) The risk of TIA/stroke recurrence is greatest within the first weeks after the index event. The outcomes in the study were among other things number of MACE within 7 and 90 days and mortality within 90 days. However, it is not clarified if the time window starts with index event or from study inclusion. Please clarify in the method section. 3) Inclusion criteria were an event (TIA or minor stroke) that had occurred within the previous month. The authors must therefore
--

	include in the manuscripte how many were included in the hyperacute phase, and how many were included later in the time course. Time from index event to study inclusion is of great importance for the results.
--	--

VERSION 3 – AUTHOR RESPONSE

Response to reviewer

Dear Dr Ihle-Hansen

Thank you for the opportunity to respond to your comments

The ProMOTE study has a pragmatic design, is nicely conducted, and has an interesting and relevant research question.

The authors responded sufficiently to my questions.

Thank you for acknowledging this. We feel that incorporating responses to your comments we have strengthened the manuscript

However, I still have some minor concerns:

1) In the conclusion section of the abstract, the authors still claim that ACR shows promise in the risk stratification of patients to avoid the admission of low-risk patients whilst assigning intensive interventions such as dual antiplatelet therapy to those at the highest risk.

The ACR is indeed a promising tool for risk stratification, but the clinical implication of the results is uncertain. Hence, the last part of the sentence should in my opinion be removed.

This latter part has been removed as suggested in the abstract (Page 4 of Marked Copy)

2) The risk of TIA/stroke recurrence is greatest within the first weeks after the index event. The outcomes in the study were among other things number of MACE within 7 and 90 days and mortality within 90 days. However, it is not clarified if the time window starts with index event or from study inclusion. Please clarify in the method section.

This has been clarified in the statistical analysis on page 10 of the marked copy. In keeping with the pragmatic nature of the study, we used the clinic appointment as the time from event rather than the index event that led to the clinic appointment.

3) Inclusion criteria were an event (TIA or minor stroke) that had occurred within the previous month. The authors must therefore include in the manuscripte how many were included in the hyperacute phase, and how many were included later in the time course. Time from index event to study inclusion is of great importance for the results.

This is indeed a very important point. In this study, 1,049 participants were seen in clinic in the first 48 hour after their index event, 83 of whom had a MACE event. Incorporating this as a covariate made no substantive difference to the model. When repeating the analysis in those who were seen within 48 hours and those who were seen later after their index event, the predictive value was numerically superior (HR 2.84 vs 2.02) however there were no significant interactions.

The time from event to clinic has been included in table 1, and a comment added on page 13.